# OpenReview forum: "Can Large Language Models Really Recognize Your Name?"
_ICLR.cc/2026/Conference — ICLR 2026 Conference Withdrawn Submission_

### Official Review · Reviewer_HMGP · 2025-10-27

**Soundness:** 2
**Presentation:** 2
**Contribution:** 2
**Rating:** 2
**Confidence:** 4

**Summary:**

The authors argue that LLMs are increasingly used in privacy-preserving systems under the implicit assumption that they can recognize sensitive information such as human names. They aim to show that LLMs struggle to correctly identify rare human names that resemble other entities. By embedding real human names that are only one edit distance away from non-human entities into short sentences, the authors demonstrate that LLMs easily confuse them with non-human entities — Name Regularity Bias. Moreover, they show that when a text unintentionally contains sentences that LLMs may misinterpret as commands, it can negatively affect privacy-preserving summarization — Benign Prompt Injection. The authors conduct extensive experiments using multiple state-of-the-art LLMs, as well as tools such as Anthropic’s Clio System and Private AI.

**Strengths:**

While Name Regularity Bias in Named Entity Recognition has been widely studied, this paper takes an interesting approach by examining Name Regularity Bias and Prompt Injection from a privacy-preserving perspective, leveraging human names that closely resemble non-human entities — particularly through a systematic collection based on an edit distance of one. The conceptualization of benign prompt injection, which is non-malicious in nature, is also a contribution. Furthermore, the construction of over 60,000 data points using LLMs and the evaluation of performance across a variety of LLMs and NER methods are notable strengths of the paper.

**Weaknesses:**

Despite its interesting aspects, this paper suffers from several **fatal flaws**.

---

### 1. Issues in Dataset Construction

The first major problem lies in the dataset creation process.
Since the primary contribution of this paper is the construction of a new benchmark, the benchmark generation pipeline is central to its value.
However, the paper merely states that five templates were selected for each name type **without describing the criteria for template selection**.
Moreover, there is no explanation of how the final human inspection was conducted.
The “manual selection” process may have inadvertently amplified the adversarial nature of the benchmark.
Inadequate manual validation could also have caused control failures in the subsequent human evaluation discussed later.

---

### 2. Poorly Defined Problem, Overclaims, and Lack of Novelty

The definitions of key problems are imprecise, the claims are overstated, and the novelty is limited.
Core concepts such as **Name Regularity Bias (NRB)**, **Prompt Injection (PI)**, and **Personally Identifiable Information (PII)** detection are already well established.
Rather than revealing new findings, this benchmark merely **reconfirms known phenomena** and does not propose practical mitigation strategies.

Although the authors frame the dataset’s contribution from a PII perspective, in reality, the benchmark only deals with **simple human names**.
Each sample consists of two very simple sentences that contain almost no personal information beyond names.
Thus, linking this benchmark to **PII performance** constitutes an overclaim, as the dataset does not provide sufficient complexity to analyze models’ PII-handling ability.

There are also issues regarding **baselines and fairness**.
While the authors state that U.S. baby *first names* were used for the baseline (line 236), the benchmark itself appears to include *surnames* (and possibly non-U.S. ones) (line 214).
If that is not the case, it should be clearly clarified.

Importantly, when both human and non-human entity names can fit grammatically into the same sentence, the task effectively becomes a **memorization problem** over diverse human names.
From an application standpoint, such a problem could be trivially solved through exact matching, undermining the benchmark’s practical relevance.

---

### 3. Methodological Limitations and Serious Control Failures (Human Evaluation)

Although the paper claims to operate under *“non-adversarial settings,”*  the name selection restricted to **edit distance 1** is inherently adversarial,  and the template validation intentionally maximizes ambiguity.  This creates a substantial distributional gap from real-world data.  (Figure 5’s frequency distribution indirectly supports this concern.)
As a result, it becomes unclear whether the observed failures are truly due to **NRB** and **BPI**,  or merely reflect the **difficulty of overly ambiguous templates**.  The paper lacks ablation studies to disentangle these effects.
In particular, the similar performance between *No BPI* and *BPI* conditions in Table 3 suggests the need for deeper analysis.
Likewise, while the *average privacy audit score* differs significantly across baselines, the *% >= score* remains small — an inconsistency that warrants explanation.

**Table 2’s control groups** reveal serious flaws in human evaluation design.
Control (*non-human*) entities are incorrectly classified as *PERSON* at 23–86% rates (average 40%), with *location* entities reaching 86% — indicating that linguistic cues (e.g., pronouns like “she”) override even obvious non-human names.
Meanwhile, control (*human*) names achieve only 55–64% recall for *syndrome/mineral* types, suggesting **severe task confusion** that invalidates these results as reliable benchmarks. These control failures imply that the templates are over-engineered to create artificial difficulty, raising fundamental doubts about whether the benchmark measures realistic privacy vulnerabilities or merely **artifacts of adversarial design**.

---

### 4. Writing and Presentation Issues

The overall flow of the paper is unnatural and repetitive. The authors should revise for conciseness to improve readability and avoid redundancy.  All figures and tables should be rewritten to fully convey their intended information without omissions.

In **Table 1**, clearly specify the criteria for boldface and include all abbreviations in full.
The term **PII** appears in the text without first introducing its full form in the main text.

**Figure 4** is especially problematic:
the caption **omits the y-axis description entirely** and fails to define the key metric of “consistency” anywhere in the paper.
The x-axis explanation (“ratio of templates in which a name is classified as human”) assumes readers already know that each name appears in exactly five templates, creating unnecessary confusion.
The vague takeaway that models are “inconsistent for at least 10%” lacks clarity — does “inconsistent” mean *consistency < 1.0*, *< 0.5*, or something else?
Moreover, this claim **contradicts the visual evidence** in subfigures (b) and (d), which show far higher inconsistency rates.
This poor presentation raises questions about whether the manually selected templates were intentionally chosen
to maximize the inconsistency effect.

Overall, Figure 4 exemplifies the broader issue throughout the paper:  critical experimental details are left to implicit understanding rather than explicit documentation.

---

**Questions:**

In addition to the weaknesses mentioned above,

did the authors consider other defense mechanisms of Clio (e.g., differential privacy in clustering)?
As recent LLM applications often improve performance through agent systems or self-refinement techniques,
does the proposed method show any performance improvement under such approaches?

---

> ### Author Response · Authors · 2025-11-23
>
> Thank you for your review. To address your concerns:
>
> ### **1. Manual template selection and data construction**
> Our manual selection step was intended solely to prevent rare semantic errors that occasionally occur during LLM-based template generation. Two annotators fluent in English independently inspected all generated templates to ensure that each template was semantically valid for both human and non-human entities. We then randomly sampled five templates per name category from those receiving unanimous approval. We will update the manuscript to include these details.
>
> ### **2. Claims, novelty, and problem framing**
> We *respectfully disagree* with the reviewer's claim that our work offers limited novelty or merely confirms known phenomena.
>
> **(a) Novel empirical finding:**
> To our knowledge, our paper is the first to demonstrate that LLMs are not reliable at detecting human names, even when the names appear explicitly and the task requires only rote extraction, under contextual ambiguity created by either Name Regularity Bias (NRB) or Benign Prompt Injection (BPI) stemming from innocuous phrasing.
>
> While NRB and prompt injection are established concepts, their privacy implications in the context of name detection by modern LLMs have not been studied. Prior privacy-preserving LLM systems (summarized in our Appendix Table 5) implicitly assume this capability is robust. Our results challenge this assumption.
>
> **(b) Scope of our benchmark:**
> We do not claim to evaluate general PII-handling performance. Instead, we characterize a specific, previously overlooked failure mode involving explicit human names, which is arguably among the simplest types of PII. Demonstrating systematic failure even on such minimal inputs is itself a significant privacy concern.
>
> We further show that similar issues arise for Reddit handles (Section 7 and Appendix D.4), suggesting broader applicability beyond conventional names. In general, any free-form PII without a fixed structure can be vulnerable to this ambiguity failure mode.
>
> **(c) Baseline fairness:**
> The reviewer notes that our baselines use U.S. first names while our benchmark includes global first and last names. This is intentional: privacy protections mediated by LLMs should not depend on whether one has a popular U.S. name or a rarer international name. Our results highlight precisely this inequity.
>
> **(d) “Exact matching” is not a valid solution:**
> Name detection cannot be reduced to memorization or exact matching. Human names are open-class, unbounded, and constantly evolving (e.g., Elon Musk’s child who is named “X Æ A-12”). No comprehensive database exists (or can exist) that enumerates all possible contemporary and future names. LLM-based privacy systems are already deployed under the assumption that they can generalize beyond known names. Our results show that this generalization is unreliable under ambiguity.

---

> > ### Author Response · Authors · 2025-11-23
> >
> > ### **3. Methodology, attribution of effects, and human evaluation**
> > **(a) “Non-adversarial settings”:**
> > Our use of the phrase refers to settings where sensitive information is explicitly present and the LLM’s task is straightforward categorization, in contrast with adversarial cases where sensitive information must be inferred or obfuscated. We will clarify this wording in the paper.
> >
> > **(b) Synthetic data and distributional shift:**
> > Synthetic templates are not a limitation for this privacy analysis. On the contrary, they allow us to systematically explore cases, specifically rare or ambiguous names, that appear sparsely in natural corpora but are crucial for equitable privacy protection. We discuss this necessity in Section 7 (lines 449–460).
> > In privacy protection, the cost of false negatives (failing to identify a person) is **substantially higher** than the cost of false positives. A robust privacy detector should not depend on high-frequency or stereotypically human contexts to succeed; otherwise, protection becomes uneven across demographic groups and rare-name populations. Our aim is not to produce unnatural sentences, but to include constructions that are plausible yet rare, which is common in real-world **long-tail privacy** cases and can promote **fairness** in privacy.
> >
> > **(c) Distinguishing NRB/BPI effects from template ambiguity:**
> > The reviewer raises a desire for ablations, but our main experiments have in fact already isolated these factors:
> >
> > - Leakage detection (Table 1): Using popular-name baselines demonstrates that failures do not arise purely due to the templates.
> >
> > - Clio evaluation (Table 3): Only ambiguous names exhibit substantial differences between No-BPI and BPI settings, while baselines remain stable. This pattern is consistent across both mean leakage and the % >= score metrics. The % >= score refers to the % of texts that the Clio rates as more private when BPI is applied than when BPI is not applied. Thus, since the average privacy rating is lower with BPI, % >= score should not increase.
> >
> > The ambiguous templates clearly didn’t have an impact by themselves, with or without BPI. To re-emphasize, the contextual ambiguity in our benchmark comes from the combination of NRB/BPI and the templates and cannot be attributed solely to the templates.
> >
> > **(d) Human evaluation “control failures”:**
> > The reviewer interprets variation in human responses as methodological failure. We believe these results instead highlight how non-experts naturally approach ambiguous privacy judgments: humans rely heavily on syntactic cues and may override their world knowledge. For example, pronouns such as “she” referring to a country are rare but valid in literary usage; it is unsurprising that non-experts interpret such cues inconsistently.
> >
> > We emphasize that the human evaluation was not used to define the benchmark or determine correctness, but only to contextualize how non-privacy-expert humans perceive these edge cases. The benchmark itself does not depend on human labels.
> >
> > ### **4. Writing and presentation**
> >
> > We appreciate the detailed editorial suggestions and will revise the manuscript accordingly (e.g., clarify definitions & figure captions, and improve the explanation of the consistency metric in Figure 4).
> >
> > Regarding the reviewer’s concern that “LLMs are inconsistent for at least 10% of names” contradicts Figure 4: the statement sets a lower bound, and a higher observed inconsistency is consistent with that interpretation. By “inconsistent”, we mean consistency < 1.0. The templates were not selected to maximize inconsistency, only to ensure naturalness and bidirectionality (usable for both human and non-human names).
> >
> > ### **5. Additional question: defenses and system integrations**
> > **(a) Clio defenses:**
> > Clio relies on prompt-based LLM judgments without additional differential privacy mechanisms. Our work shows that such reliance is insufficient, even within a leading industry system. Incorporating DP-based mechanisms could help, but designing a comprehensive defense is beyond our scope.
> >
> > **(b) Agentic or self-refining LLMs:**
> > Agent systems ultimately rely on LLMs for name detection and reasoning; no realistic non-AI tool can reliably identify sensitive names given their open-class nature. Self-refinement could improve performance, as suggested by our Figure 2, but developing such systems is outside the scope of this work. Our benchmark is intended as a guardrail for evaluating privacy vulnerabilities, not as training data.
> >
> > We appreciate the reviewer’s feedback and believe that the clarifications above have addressed the concerns raised. We look forward to hearing back from the reviewer.

---

> > > ### Comment · Reviewer_HMGP · 2025-11-27
> > >
> > > I appreciate the effort the authors put into the response; however, my view remains unchanged.
> > >
> > > First, I would like to see the revisions properly reflected in the updated manuscript rather than only in the rebuttal.
> > >
> > > Your response does not sufficiently address several key concerns I raised. Among them, the most critical issues are as follows:
> > >
> > > You did not provide a convincing counterargument regarding why this experimental setting should not be considered a methodological failure. You claim that the human evaluation is intended to *“highlight how non-experts naturally approach ambiguous privacy judgments,”* yet no supporting justification or evidence is provided. As not only I but also Reviewer **B2Qw** pointed out, *“the evaluated setting may not accurately reflect practical privacy risks, but rather test linguistic robustness under artificially ambiguous cases.”*
> > >
> > > Although the paper is framed from the perspective of PII, the work deals only with personal names. There is no discussion on how the proposed approach could generalize beyond names or evolve toward other forms of PII. A major restructuring of the narrative throughout the manuscript appears necessary, and this issue was also noted by Reviewer **4axf**.

---

> > > > ### Author Response · Authors · 2025-11-30
> > > >
> > > > Thank you for your response, we really appreciate your continued engagement. We regret not being able to continue to interact meaningfully with the reviewer, so we hope our explanation below will help address your concerns.
> > > >
> > > > ### **1. Human evaluation**
> > > >
> > > > Regarding the human evaluation, we'd like to invite the reviewer to inspect some actual example templates used in our evaluation and human study (the full list can be found in Appendix B.2):
> > > >
> > > > Starting with two templates for **location-like** names:
> > > > - "I have always relied on [MASK] for support. She offers more than I could ever ask for."
> > > > - "I am slowly getting to know [MASK] better. She feels both familiar and mysterious at the same time."
> > > >
> > > > Without any names, the majority of humans will likely consider these templates to be talking about a person at first glance. Only a very small number of participants in our study were actually aware of the use of the pronoun "she" to refer to a country (according to the post-survey responses). As a result, even when [MASK] is replaced with a known country name like "Germany" or "Ethiopia", the majority of participants still relied on their understanding of the templates without the name to guess the entity type. However, what should be the ground truth in this case? There are actually some humans with names exactly like "Germany" or "Ethiopia" as well. The difference between these and the names in AMBENCH is that the latter are validated to be exclusive to humans, meaning no other non-human entities would bear those names. As such, the ground truth for the latter is very clearly humans. Thus, the human results for the non-human control samples for the location category do not reflect the ground truths, but rather the humans' prior beliefs and knowledge (e.g., "she" as a stylistic personalization of countries, ships, etc.).
> > > >
> > > > Here are two other templates for the **mineral-like** category:
> > > > - "I enjoy sharing knowledge about [MASK]. There is something about the subject that sparks curiosity and fascination."
> > > > - "I recently read about [MASK]. The subject has a reputation for being deceptive at first glance."
> > > >
> > > > The phrase "the subject" can be used to refer to a person or a thing/concept, but whether it's more aligned with a person or a thing can depend on the context and also on the reader's perception. Students (who form the majority of our respondents) might perceive "subject" more as a thing/concept (e.g., a topic to be studied in a class), but a researcher who reads a lot of human studies might perceive it as "human subjects".
> > > >
> > > > Overall, we are confident our methodology is sound, but our explanations of the results can be extended. We'll conduct additional human evaluation to study their perception of the names and templates in a more controlled setting. At the same time, we'd also like to ask the reviewer to assess whether these templates look adversarial. From the manual annotators' and also GPT-4o's assessments, these short snippets look natural. It might be argued that they are too short, but should a privacy-preserving solution only work for longer contexts? This also touches on the **fairness** issue: our benchmark may be artificial, but with the lack of real data to ensure fairness in ML, synthetic edge cases can help remediate this issue.
> > > >
> > > > ### **2. Extensibility to other PII types**
> > > >
> > > > We have already discussed how our approach can be extended to other PII types in Section 7 and Appendix D.4. To reiterate, **any free-form PII without a fixed structure**, such as location or organization names, can be vulnerable to the ambiguity failure mode identified in our paper. We demonstrated this with more than 4000 real Reddit handles (e.g., too-old-for-this, its-tough-outthere, dont-cancel-my-score, panda-with-a-hug, show-me-the-honey). Fixed-format PIIs like emails, phone numbers, etc., are not vulnerable to this issue.

---

### Official Review · Reviewer_4axf · 2025-10-31

**Soundness:** 3
**Presentation:** 3
**Contribution:** 2
**Rating:** 4
**Confidence:** 4

**Summary:**

This paper explores how efficiently a variety of LLMs are capable of recognizing human names in the provided prompt context. Authors also proposed a dataset benchmark to evaluate the fact under the hood of privacy-preserving LLMs.

**Strengths:**

1.	This paper investigates a compelling issue within LLMs, efficient PII understanding in user interactions, specifically names, as well as the privacy implications under this.
2.	Proposed AmBench benchmark dataset with 60K data points, which consists of both real-world and synthetically generated scenarios.

**Weaknesses:**

1.	The paper lacks an explicit description of the problem and the use cases of the solution provided at the beginning of the paper. Thus, the readability lacks its quality. Describing a hypothetical scenario where the problem occurs and how the proposed solution may address the issue would also help.
2.	The problem setting is highly superficial. Does not provide a clear idea about the problem in a real-world environment.
3.	This paper takes only one category of PII, i.e., names, into account for analyzing the issue. Does this PII misclassification/ misidentification issue exist in other PIIs, e.g., email, phone number? If so, to what extent?
4.	There is no solid evidence/ study/research outcomes provided behind the claims in lines 165-168?
5.	Why did not the authors consider the o1 or o3 reasoning models in the evaluation?
6.	The assumption made in the attack scenario stands on some assumptions which are somewhat unrealistic, e.g., the user’s conversation is included in Clio’s inputs. If the entire conversation is unveiled to the attacker, then he can directly refer to the conversation for names; no need to perform an attack.
7.	What are the potential solutions to this issue presented in the paper?
8.	The entire paper explores the vulnerabilities of SOTA LLMs in not efficiently identifying the names in the prompt context. What's the explicit/implicit relation of it with privacy leakage? It is not explained in a clear way.
9.	“LLMs are much worse at detecting ambiguous human names than popular ones.” The explanation of popular exists nowhere. Is it popular LLMs or popular names?
10.	In the abstract, line 20 should be explained in more detail on how the proposed dataset’s names are four times more likely to be ignored in the LLM-enabled privacy-preserving tools.

**Questions:**

Please follow the Weaknesses.

---

> ### Author Response · Authors · 2025-11-23
>
> Thank you for your review. To address your concerns:
>
> ### **1 & 2. Problem description and setting**
> The core issue we study is the reliability of LLMs when used to detect sensitive information, especially human names, in unstructured text. This task has become increasingly common in real-world privacy-preserving workflows. For example, before releasing a medical dataset for research, practitioners must ensure that no residual patient PII (including names) remains in clinical notes. LLMs are attractive for this task because they handle unstructured language well.
>
> However, our results show that these systems can systematically fail when names are uncommon or appear in contexts that are linguistically ambiguous. AMBENCH is designed precisely to expose and quantify this gap, and our results demonstrate that even state-of-the-art models can miss many such names. We will add a clearer motivating scenario and an explicit early statement of the problem in the revision.
>
> ### **3. Are other PIIs affected?**
> Section 7 (“Extensibility of AMBENCH”) discusses this, but we will make the connection more explicit. Our analysis focuses on free-form PIIs, i.e., information that has no fixed structure. Names are the most prominent example, but other free-form PIIs (e.g., affiliations, locations, or usernames) can exhibit the same ambiguity-related failure modes. To demonstrate this, we also evaluated over 4,000 Reddit account handles and found that even the strongest models miss a significant fraction (Section 7). In contrast, structured PIIs such as phone numbers and emails are not susceptible to this type of ambiguity.
>
> ### **4. Evidence behind linguistic example in lines 165-168**
> The statements in lines 165-168 are not claims needing empirical evidence but linguistic observations illustrating plausible reasons LLMs might misinterpret the name Adomite (e.g., similarity to mineral names like Adamite). We’ll clarify that this is a qualitative explanation to help readers understand the failure mode, not a quantitative claim.
>
> ### **5. Why not evaluate o1/o3 reasoning models?**
> We tested GPT-5 Thinking mode, which is the successor of o1 and o3 and has significantly better reasoning performance than both of those while at a lower cost. Because GPT-5 Thinking subsumes these models, additional experiments with o1/o3 would not change the conclusions. Performance data for GPT-5 can be found at https://openai.com/index/introducing-gpt-5/
>
> ### **6. Realism of the Clio threat model**
> Clio is an LLM-powered privacy-preserving system that processes Claude users’ conversations and surfaces only sanitized, high-level insights to employees. Thus, only Clio, not Anthropic’s employees, has access to raw conversations. The attacker in our model is a malicious employee who only sees Clio’s outputs, not the original text. Therefore, the assumption that user conversations appear in Clio’s inputs is realistic and consistent with Anthropic’s own description of the system. Our results show that Clio may fail to sanitize ambiguous names, creating an opportunity for privacy leakage.
>
> ### **7. Potential solutions to ambiguity**
> Section 7 (lines 461–469) outlines two mitigation strategies, which we will highlight more clearly:
>
> (1) Improving reasoning ability, e.g., by fine-tuning models for privacy tasks. This can potentially yield higher recall but increases inference cost.
>
> (2) Providing explicit ambiguity-handling instructions, which improved performance in Appendix D.5 but requires foreknowledge of the types of ambiguous cases likely to occur.
>
> We emphasize that both mitigations can reduce but do not eliminate the underlying issue.
>
> ### **8. Connection to privacy leakage**
> The task we examine (accurate detection of sensitive names) is central to compliance workflows (e.g., HIPAA), biomedical dataset de-identification, and emerging LLM-user-safety applications such as anonymizing chat logs or social media posts. A systematic failure to detect ambiguous names directly increases the risk of privacy leakage in these settings. Section 2 surveys these use cases, but we will strengthen the framing.
>
> ### **9. Popular LLMs or popular names**
> “Popular” refers to the 200 most common U.S. baby names, which serve as our baseline for evaluating name regularity. We will make this definition explicit at its first mention. We’ll make this phrasing clearer in the manuscript.
>
> ### **10. Clarifying Abstract line 20**
> We will expand the explanation in the abstract. AMBENCH names are four times more likely to be missed by the Clio’s LLM-based auditor due to (i) the Name Regularity Bias documented in our experiments and (ii) the Benign Prompt Injection effect, which collectively cause Clio’s underlying LLM to overlook rare or ambiguous names.
>
> We believe the clarifications above have addressed the reviewer’s concern. We look forward to hearing the reviewer’s thoughts.

---

### Official Review · Reviewer_B2Qw · 2025-11-01

**Soundness:** 2
**Presentation:** 3
**Contribution:** 2
**Rating:** 4
**Confidence:** 4

**Summary:**

This paper introduces AMBENCH, a benchmark designed to evaluate large language models’ ability to recognize ambiguous human names that closely resemble non-human entities. The dataset contains over 60K samples, covering human–nonhuman name pairs like Albanir–Albania and Versache–Versace, embedded in short ambiguous sentences. It systematically tests 12 different LLMs, revealing biases in name recognition and providing valuable insights into LLMs’ limitations in privacy protection and anonymization.

**Strengths:**

1. The paper clearly describes the benchmark construction process, offering valuable insights and methodological guidance for future benchmark design and related research on LLM safety and privacy.

2. The paper conducts a systematic and comprehensive evaluation across 12 large language models, covering both reasoning and non-reasoning, as well as open-source and closed-source models.

3. The study highlights an underexplored vulnerability—LLMs’ difficulty in recognizing ambiguous personal names, which provides a new perspective for understanding the limitations of LLM-based privacy protection and anonymization systems.

**Weaknesses:**

1. While the paper introduces an interesting benchmark for testing ambiguity in name recognition, the constructed scenarios appear somewhat unrealistic. In real-world contexts, human names rarely resemble non-human entities to such a degree (just one letter difference), and sentences where both usages are plausible are uncommon. As a result, the evaluated setting may not accurately reflect practical privacy risks, but rather test linguistic robustness under artificially ambiguous cases.

2. The benchmark’s scope is quite limited, as it only considers cases where human names are nearly identical to non-human entities. Such situations are rare in real-world applications. Consequently, the benchmark may have limited practical use, as few privacy or safety systems would realistically need to be evaluated under these highly specific conditions.

3. The reported issue appears less severe for state-of-the-art reasoning models. As shown in Table 1, reasoning-capable models (e.g., GPT-5-mini, Gemini 2.5, Claude 3.5) achieve substantially higher recall and lower false discovery rates than smaller or non-reasoning models. This suggests that the identified vulnerability may diminish as LLM reasoning abilities improve, limiting the long-term significance of the benchmark’s findings.

**Questions:**

Please see the weakness above.

---

> ### Author Response · Authors · 2025-11-23
>
> Thank you for your review. To address your concerns:
>
> ### **1 & 2. Practicality and scope of AMBENCH**
>
> We argue, to the contrary, that privacy- and safety-critical systems powered by AI should be *especially* tested under uncommon scenarios. A useful example is self-driving cars: it would be inadequate and irresponsible to test such technology only under typical road conditions and then claim strong performance. Many real-world failures arise precisely because the system had not been stress-tested on **unusual yet plausible scenarios** [1]. While privacy leakage is certainly not physically life-threatening, the reputational, psychological, and socioeconomic harms it can inflict are substantial. Our benchmark is thus intended to provide this kind of rigorous stress test for privacy-preserving NLP systems.
>
> Moreover, focusing solely on common names would **undermine fairness**. Individuals with rare, cross-linguistic, or culturally marginalized names deserve the same level of privacy protection as people with common names. Yet, our results show that current LLMs underperform in these cases. In that sense, AMBENCH fills an important gap by **highlighting inequities** that would otherwise remain unnoticed if benchmarks only included frequent or standard name forms. We believe that far from being a narrow or contrived setting, our benchmark highlights an important weakness: privacy systems that silently fail for the people **least represented** in training data.
>
> All in all, AMBENCH offers both a robustness and a fairness test. We therefore believe it has practical value for developers and researchers who want to build LLM-based privacy technologies that are reliable and also equitable across the full range of human names.
>
> ### **3. Future capability of LLMs**
>
> We do not believe that the potential improved performance of LLMs should be considered as a weakness of our benchmark, or any LLM benchmark in general. The current capability of large reasoning models, while stronger than other LLM types on AMBENCH, still leaves plenty of room for improvement. Our benchmark thus supports evaluating and tracking progress toward genuinely reliable privacy reasoning. Furthermore, it can also help accelerate the process of building truly robust privacy solutions. Users should not have to wait for LLM providers to improve their LLMs before being able to protect their data.
>
> We believe our response above has addressed the reviewer’s concern. We look forward to hearing the reviewer’s thoughts on our response.
>
> ### **References**
>
> [1] List of Tesla Autopilot crashes. https://en.wikipedia.org/wiki/List_of_Tesla_Autopilot_crashes

---

### Official Review · Reviewer_rgF6 · 2025-11-03

**Soundness:** 3
**Presentation:** 3
**Contribution:** 3
**Rating:** 4
**Confidence:** 3

**Summary:**

This paper examines how large language models (LLMs) handle non-adversarial contextual ambiguity in privacy-sensitive tasks. It identifies two factors—Name Regularity Bias (NRB) and Benign Prompt Injection (BPI)—and introduces AMBENCH, a benchmark of 12K human names orthographically or semantically similar to non-human entities. Evaluating 12 LLMs and commercial PII tools, the authors find a 20–40% recall drop on ambiguous cases, revealing persistent failures in LLM-based privacy detection systems.

**Strengths:**

- The paper is well structured, easy to follow, and clearly motivated.
- The experimental setup is comprehensive: the benchmark contains ~60K sentences and covers a wide range of LLMs and commercial tools, ensuring generalizability.
- Introducing the notion of non-adversarial ambiguity as a distinct failure mode in LLM privacy detection is novel and practically meaningful.

**Weaknesses:**

1. Semantic implausibility in "ambiguous" examples

Several examples presented in the paper appear syntactically valid but semantically implausible for human agents. These constructions make it difficult to disentangle whether model errors stem from representational bias or from the unnaturalness of the input sentences themselves. For instance:
- "I managed to find traces of Adomite at the work site." (Section 3) – The phrase "find traces of" usually applies to physical substances, not people.

These cases suggest that what the authors call contextual ambiguity may partly arise from semantic mismatch in the generated templates, rather than from genuine ambiguity in model reasoning.
Furthermore, while the authors justify such constructions as part of a “worst-case evaluation” (Section 7, Adversariality of AMBENCH), a legitimate worst-case benchmark should still preserve semantic plausibility. Evaluating models on linguistically implausible or pragmatically inconsistent inputs (e.g., “find traces of Adomite”) may conflate privacy reasoning failure with limitations in general language understanding. If real-world privacy systems were tuned to handle such implausible edge cases, their performance on normal data distributions would likely degrade severely due to excessive over-scrubbing and false positives.
This may weaken the benchmark’s construct validity, as it shifts the evaluation focus from assessing privacy reasoning to robustness under linguistically implausible inputs.

2. Ambiguity defined purely by orthographic similarity

The construction of ambiguous names relies solely on Levenshtein distance ≤ 1 between human and non-human entities (Section 4, Appendix B.1). This criterion captures surface-level resemblance but not semantic or pragmatic ambiguity. Consequently, many sentences resemble artificially mixed contexts where a human name is inserted into an otherwise non-human frame.

3. Manual template selection lacks transparency

Five templates per domain were "manually selected" (Appendix B.2), but the paper does not specify how ambiguity was judged or whether multiple annotators were involved. The absence of inter-annotator agreement raises concerns about reproducibility and objectivity of AMBENCH’s construction.

**Questions:**

- Some “ambiguous” examples (e.g., “find traces of Adomite”, “responsible for the infection”) appear semantically incompatible with human subjects. On what basis were these cases considered ambiguous? Did you validate their naturalness through human annotation or any linguistic plausibility check?
- Since ambiguity is defined by Levenshtein distance ≤ 1, how do you distinguish genuine contextual ambiguity from mere lexical similarity? Would you consider adding a contrastive setup (e.g., placing the same names in clearly human-centric contexts) or linguistic plausibility metrics (e.g., selectional preference or perplexity) to validate semantic naturalness?
- The five domain templates were manually selected (Appendix B.2). Could you elaborate on the selection process and whether inter-annotator agreement was measured to ensure reproducibility?

---

> ### Author Response · Authors · 2025-11-23
>
> Thank you for your review. To address your concerns:
>
> ### **1. Semantic plausibility**
> We make sure the templates are semantically plausible for **both** human and non-human names as part of the benchmark data generation pipeline itself (Figure 3). Specifically, we prompt GPT-4o to assess the naturalness of the templates for both types of names, then we filter for templates that are considered valid by GPT-4o. We also involve two human annotators to manually verify the semantic plausibility of the filtered templates (more details about this manual process below).
>
> In our motivational example in Section 3 ("I managed to find traces of <ENTITY> at the work site."), we acknowledge that the phrase “find traces of” might be more common for substances, but it is nonetheless plausible for human entities as well (e.g. “find traces of Picasso” [1]). An ideal LLM should be able to reason about these possibilities without discarding names that may merely look non-human.
>
> Finally, in privacy protection, the cost of false negatives (failing to identify a person) is **substantially higher** than the cost of false positives. A robust privacy detector should not depend on high-frequency or stereotypically human contexts to succeed; otherwise, protection becomes uneven across demographic groups and rare-name populations. Our aim is not to produce unnatural sentences, but to include constructions that are plausible yet rare, which is common in real-world **long-tail privacy** cases and can promote **fairness** in privacy.
>
> ### **2. Contextual ambiguity vs lexical similarity**
> We appreciate the concern that Levenshtein distance alone may fail to capture true ambiguity. To clarify: we **do not** define contextual ambiguity solely by orthographic similarity. Ambiguity emerges from the interaction between a name that is orthographically similar to a non-human term and a template that is semantically plausible for both human and non-human interpretations.
>
> To empirically isolate the effect of lexical similarity alone, we conducted an additional experiment. We modified the second sentence of each template to be overtly human-centric by changing how we referred to the entity (i.e., replacing “the manager…” with “He…”), then reevaluated some representative LLMs as well as Flair. The results in the table below (Recall / False Discovery Percentage) show that all methods now classify the ambiguous name just as reliably as the baseline popular human names once the context is clearly human-oriented. This demonstrates that the models’ errors in AMBENCH stem from contextual ambiguity, not from orthographic similarity alone.
>
> |Method|Location|Organization|Syndrome|Mineral|Bacteria|Baseline|
> |----------|-----------|-----------|-----------|-----------|-----------|-----------|
> |Gemini 2.5 Flash|0.995 / 0.2|0.976 / 4.7|0.994 / 0.2|0.975 / 0.04|0.989 / 0.09|1.0 / 0.1|
> |Gemma 2 9B|0.998 / 0.2|0.993 / 5.6|0.996 / 0.3|0.995 / 0.3|0.989 / 1.0|0.999 / 2.6|
> |Flair|0.95 / 0.0|0.990 / 0.0|0.994 / 0.0|0.997 / 0.0|0.997 / 0.0|0.995 / 0.0|
>
> Regarding linguistic plausibility metrics like perplexity or selectional preference, because the ambiguous names in our benchmark are low-frequency, those metrics would be confounded by rarity rather than mismatch. However, we re-emphasize that in the context of privacy, it’s imperative that we test privacy solutions on less common cases to ensure fair privacy protection.
>
> ### **3. Manual template selection**
> This step is primarily to prevent any semantic issue in the LLM-generated templates that was not caught by the self-validation stage, which happens fairly rarely, mostly due to the templates using a phrase that is exclusive to either human or non-human entities. Two annotators fluent in English assessed the generated templates on whether they are semantically sound for both human and non-human entities. Afterward, for each name category, we randomly selected 5 templates where both annotators agreed on to construct the final benchmark. Inter-annotator agreement (Cohen’s Kappa) was above 0.8, indicating strong agreement. We will update the manuscript with details about this step.
>
> We believe the clarifications above have addressed the reviewer’s concern. We look forward to hearing the reviewer’s thoughts.
>
> ### **References**
>
> [1] Some examples of the phrase “find traces of” being applied to human names: ["find traces of Picasso"](https://koha.mk/en/malaga-qyteti-i-mrekullueshem-spanjoll-ku-vizitoret-gjejne-gjurmet-e-picassos-kudo), ["find traces of Alice"](https://www.lrb.co.uk/the-paper/v30/n01/colm-toibin/a-man-with-my-trouble)

---

### Note · Authors · 2026-01-05

I have read and agree with the venue's withdrawal policy on behalf of myself and my co-authors.